# Diabetes medications and associations with Covid-19 outcomes in the N3C database: A national retrospective cohort study

Carolyn T. Bramante[1]*, Steven G. Johnson[2], Victor Garcia[3], Michael D. Evans[4], Jeremy Harper[5], Kenneth J. Wilkins[6], Jared D. Huling[7], Hemalkumar Mehta[8], Caleb Alexander[9], Jena Tronieri[10], Stephenie Hong[8], Anna Kahkoska[11], Joy Alamgir[12], Farrukh Koraishy[13], Katrina Hartman[1], Kaifeng Yang[7], Trine Abrahamsen[14], Til Stürmer[15], John B. Buse[16], N3C core authors¶

1 Division of General Internal Medicine, Department of Medicine, University of Minnesota Medical School, Minneapolis, Minnesota, United States of America, 2 Institute for Health Informatics, University of Minnesota Medical School, Minneapolis, Minnesota, United States of America, 3 Department of Biomedical Informatics, Stony Brook University Hospital, Stony Brook, New York, United States of America, 4 Biostatistical Design and Analysis Center, University of Minnesota Medical School, Minneapolis, Minnesota, United States of America, 5 Owl HealthWorks, Indianapolis, IN, United States of America, 6 Biostatistics Program, Office of the Director, National Institute of Diabetes and Digestive and Kidney Disease, Bethesda, Maryland, United States of America, 7 Division of Biostatistics, University of Minnesota School of Public Health, Minneapolis, Minnesota, United States of America, 8 Division of Epidemiology and Methodology, Johns Hopkins School of Public Health, Baltimore, Maryland, United States of America, 9 Division of General Internal Medicine, Department of Medicine, Johns Hopkins School of Medicine, Baltimore, Maryland, United States of America, 10 Department of Psychiatry, Perelman School of Medicine at the University of Pennsylvania, Philadelphia, Pennsylvania, United States of America, 11 Department of Nutrition, Gillings School of Global Public Health, University of North Carolina at Chapel Hill, Chapel Hill, North Carolina, United States of America, 12 ARIScience, Boston, Massachusetts, United States of America, 13 Division of Nephrology, Stony Brook University Hospital, Stony Brook, New York, United States of America, 14 Novo Nordisk, Bagsvaerd, Denmark, 15 Department of Epidemiology, Gillings School of Global Public Health, University of North Carolina at Chapel Hill, Chapel Hill, North Carolina, United States of America, 16 Division of Endocrinology, Department of Medicine, University of North Carolina Medical School, Chapel Hill, North Carolina, United States of America

¶ Membership of the N3C core authors is listed in the Acknowledgments.
* bramante@umn.edu

**Data Availability Statement:** The NIH has ethical and legal restrictions on the data because it contains sensitive patient information. The data is

## Abstract

### Background

While vaccination is the most important way to combat the SARS-CoV-2 pandemic, there may still be a need for early outpatient treatment that is safe, inexpensive, and currently widely available in parts of the world that do not have access to the vaccine. There are in-silico, in-vitro, and in-tissue data suggesting that metformin inhibits the viral life cycle, as well as observational data suggesting that metformin use before infection with SARS-CoV2 is associated with less severe COVID-19. Previous observational analyses from single-center cohorts have been limited by size.

### Methods

Conducted a retrospective cohort analysis in adults with type 2 diabetes (T2DM) for associations between metformin use and COVID-19 outcomes with an active comparator design of

available in the N3C public repository. The data is available with approval, more information can be found here: https://ncats.nih.gov/n3c/resources/data-access. Contact information for data access requests: NCATSPartnerships@mail.nih.gov.

**Funding:** The analyses described in this publication were conducted with data or tools accessed through the National Institutes of Health's National Center for Advancing Translational Sciences (NCATS) N3C Data Enclave (https://covid.cd2h.org) and N3C Attribution & Publication Policy v 1.2-2020-08-25b supported by NCATS U24 TR002306 and NCATS grants KL2TR002492 (CB) and UL1TR002494 (UMN, CB and SJ) and UL1TR002489 (UNC, JB); and and the National Institute of Digestive, Diabetes, and Kidney diseases K23DK124654 (CB), and K23DK116935 (JT). The content is solely the responsibility of the authors and does not necessarily represent the official views of the National Institutes of Health's National Center for Advancing Translational Sciences. TS receives investigator-initiated research funding and support as Principal Investigator (R01 AG056479) from the National Institute on Aging (NIA), and as Co-Investigator (R01 HL118255, R01MD011680), National Institutes of Health (NIH). He also receives salary support as Director of Comparative Effectiveness Research (CER), NC TraCS Institute, UNC Clinical and Translational Science Award (UL1TR002489), the Center for Pharmacoepidemiology (current members: GlaxoSmithKline, UCB BioSciences, Takeda, AbbVie, Boehringer Ingelheim), from pharmaceutical companies (Novo Nordisk), and from a generous contribution from Dr. Nancy A. Dreyer to the Department of Epidemiology, University of North Carolina at Chapel Hill. Dr. Stürmer does not accept personal compensation of any kind from any pharmaceutical company. He owns stock in Novartis, Roche, and Novo Nordisk.

**Competing interests:** I have read the journal's policy and the authors of this manuscript have the following competing interests: TS owns stock in Novartis, Roche, and Novo Nordisk. T.J.A. is an employee of Novo Nordisk and reports personal fees and non-financial support from Novo Nordisk during the conduct of the study, as well as personal fees from Novo Nordisk outside the submitted work. J.A. is founder of ARIScience. J.B. B.'s contracted consulting fees and travel support for contracted activities are paid to the University of North Carolina by Adocia, AstraZeneca, Eli Lilly, Intarcia Therapeutics, MannKind, Novo Nordisk, Sanofi, Senseonics, and vTv Therapeutics; he reports grant support from AstraZeneca, Dexcom, Eli Lilly, Intarcia Therapeutics, Johnson & Johnson,

prevalent users of therapeutically equivalent diabetes monotherapy: metformin versus dipeptidyl-peptidase-4-inhibitors (DPP4i) and sulfonylureas (SU). This took place in the National COVID Cohort Collaborative (N3C) longitudinal U.S. cohort of adults with +SARS-CoV-2 result between January 1 2020 to June 1 2021. Findings included hospitalization or ventilation or mortality from COVID-19. Back pain was assessed as a negative control outcome.

## Results

6,626 adults with T2DM and +SARS-CoV-2 from 36 sites. Mean age was 60.7 +/- 12.0 years; 48.7% male; 56.7% White, 21.9% Black, 3.5% Asian, and 16.7% Latinx. Mean BMI was 34.1 +/- 7.8kg/m$^2$. Overall 14.5% of the sample was hospitalized; 1.5% received mechanical ventilation; and 1.8% died. In adjusted outcomes, compared to DPP4i, metformin had non-significant associations with reduced need for ventilation (RR 0.68, 0.32–1.44), and mortality (RR 0.82, 0.41–1.64). Compared to SU, metformin was associated with a lower risk of ventilation (RR 0.5, 95% CI 0.28–0.98, p = 0.044) and mortality (RR 0.56, 95% CI 0.33–0.97, p = 0.037). There was no difference in unadjusted or adjusted results of the negative control.

## Conclusions

There were clinically significant associations between metformin use and less severe COVID-19 compared to SU, but not compared to DPP4i. New-user studies and randomized trials are needed to assess early outpatient treatment and post-exposure prophylaxis with therapeutics that are safe in adults, children, pregnancy and available worldwide.

## Introduction

The novel severe acute respiratory syndrome coronavirus 2 (SARS-CoV-2) continues to spread globally and evolve into variants that may be more infectious and may evade current vaccines and therapies [1]. While vaccine development and distribution remains the primary way to combat the COVID-19 pandemic, many individuals around the world do not yet have access to these vaccines, young children are not yet vaccinated, and large percentages of those with access are not willing to be vaccinated [2, 3]. Thus there appears to be a need for early outpatient treatment options that are safe, inexpensive, and widely available to prevent severe symptoms, hospitalization, critical illness, and mortality associated with SARS-CoV-2 infection.

To this end, several medications have been suggested for repurposing to treatment of SARS-CoV-2 [4]. Of these, metformin seemed to warrant further investigation given its widespread use in adults, children, pregnancy, and its availability worldwide for less than $2 per month [5–9]. Metformin is known to inhibit mTOR (mechanistic target of rapamycin), which appears to be important for replication of SARS-CoV-2 [10, 11]. Metformin has been shown to inhibit the viral life cycle of other RNA viruses [12]. Beyond affecting the viral life cycle, metformin has anti-inflammatory and anti-thrombotic properties, which may also reduce severity of COVID-19 disease [13–15]. In addition, there are in-vitro, in-silico, and observational data suggesting that metformin use may reduce the severity of COVID-19 disease [10, 11, 16–18]. However, observational analyses are limited because of confounding by indication,

Lexicon, NovaTarg, Novo Nordisk, Sanofi, Theracos, Tolerion, and vTv Therapeutics; he has received fees for consultation from Anji Pharmaceuticals, AstraZeneca, Boehringer Ingelheim, Cirius Therapeutics Inc, Eli Lilly, Fortress Biotech, Janssen, Mellitus Health, Moderna, Pendulum Therapeutics, Praetego, Stability Health, and Zealand Pharma; he holds stock/options in Mellitus Health, Pendulum Therapeutics, PhaseBio, Praetego, and Stability Health; and he is supported by grants from the National Institutes of Health, Patient Centered Outcomes Research Institute, Juvenile Diabetes Research Foundation International and the American Diabetes Association. Dr. Bramante holds an FDA IND for investigation of metformin for early outpatient treatment of COVID-19, NCT04510194. This does not alter our adherence to PLOS ONE policies on sharing data and materials.

which is particularly relevant for metformin because diabetes is a risk factor for poor outcomes from COVID-19. One pharmaco-epidemiologic approach to assess potential pleiotropic effects of medications while minimizing confounding by indication is to compare individuals with the same condition (the same indication), and same engagement in healthcare (taking similarly-available medications), on therapeutically equivalent medications.

From a type 2 diabetes (T2DM) treatment standpoint, metformin, Dipeptidyl peptidase 4 inhibitors (DPP4i), and sulfonylureas (SU's) are therapeutically equivalent. Thus, comparing individuals with type 2 diabetes treated with monotherapy of one of these three medications may reduce confounding by indication, which is important given that diabetes is a significant risk factor for poor outcomes from COVID-19. DPP4i have been hypothesized to reduce severity of COVID-19 disease, by reducing viral entry into the cell and DPP4i's have also been associated with reduced inflammation, and blocking viral entry has not been a strong pathway for stopping the virus in other medications [19–21]. Metformin has more favorable profile regarding cost and medication interactions, so it is important to understand whether it would offer benefit compared to DPP4i's. Sulfonylureas have no hypothesized benefit in SARS-CoV-2 infection beyond treatment of pre-existing diabetes in individuals with comorbidities. Comparing metformin to SU's is important for understanding if metformin offers any benefit beyond treating diabetes.

Previous observational analyses assessing metformin and COVID-19 outcomes have had limitations such as geographic homogeneity, lack of BMI data, and insufficient numbers for comparing diabetes monotherapy groups [17, 18, 22–24]. Our objective was to address some of these limitations by using the N3C database (National COVID Cohort Collaborative), a large nationally representative dataset of electronic health record (EHR) data, [25] to assess COVID-19 outcomes in adults with type 2 diabetes (T2DM). We used an active comparator design of prevalent users of diabetes monotherapy: metformin versus sulfonylureas (SU) and DPP4i. We hypothesized that metformin use prior to SARS-CoV-2 infection would be associated with less severe COVID-19 outcomes than SU and DPP4i use.

## Methods

### Design and population

We performed a retrospective cohort analysis of patient-level, de-identified EHR data from 2017 to May 2021. The N3C includes data from 56 institutions nationally, across geographically and diverse areas [25]. This analysis was approved by the University of Minnesota institutional review board (STUDY00011578), which provided a waiver of consent. We used an active comparator prevalent user design of diabetes monotherapy with either metformin, SU, or DPP4i.

### Inclusion and exclusion criteria

The dataset included 1.6 million individuals with a positive SARS-CoV-2 polymerase chain reaction (PCR) result between 1/1/2020 to 12/12/2020 (Fig 1), with EHR records extending back two years for medical histories. Analysis was restricted to adults over age 30 years with T2DM and at least 1 outpatient healthcare encounter in the 12 months before the +- SARS-CoV-2 result. This age minimum was chosen to enrich the population as 30 is the age at which the risk of hospitalization appears to rise above 5% [26]. T2DM was defined as having at least one diabetes pharmacotherapy agent and either a hemoglobin A1C (HbA1C) level > = 6.5% or an ICD-10 code for diabetes in the previous 12 months. To reduce confounding by contraindication, individuals were excluded if they had a diagnosis of chronic kidney disease (CKD) Stage 4, Stage 5, or End Stage Renal Disease (ESRD). Records of individuals with

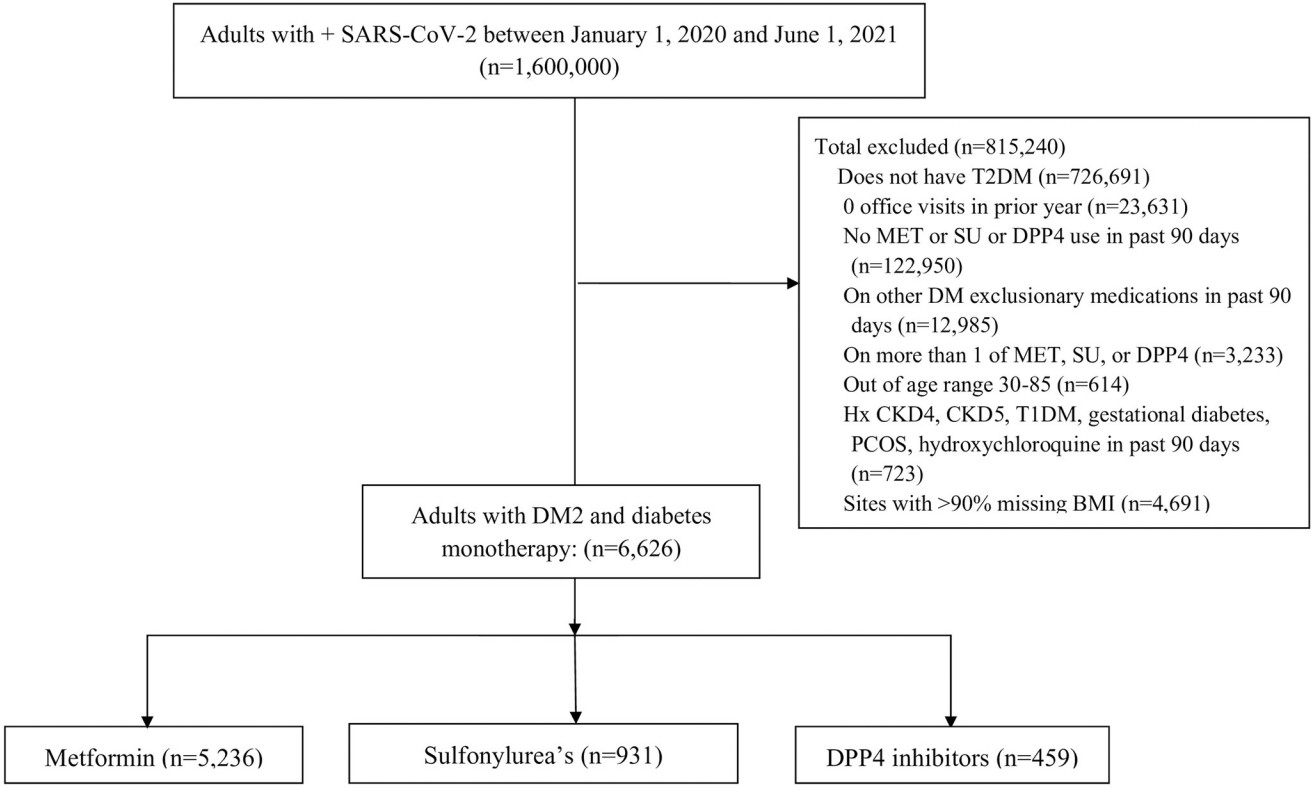

**Fig 1. This represents the number of patients included in the analysis after the inclusion and exclusion criteria were applied.**

prediabetes or polycystic ovarian syndrome (two common uses for metformin other than T2DM), but not T2DM, were excluded. To reduce confounding by frailty, individuals over age 85 years were excluded.

## Exposure groups

Metformin, SU, and DPP4i use was determined by being reported on the patients' active medication list within the 90 days prior to the +SARS-CoV-2 result. Using both the WHO ATC classification and RxNorm schemas, concept sets were created for the drugs of interest. A 2-physician review team manually read through each concept expression to assure appropriate inclusion of concepts to the expression list. Individuals were excluded from the analysis if any other diabetes medication were listed in the 90 days prior to the SARS-CoV-2 result.

## Outcomes

The clinical outcomes of interest were hospital admission for COVID-19 disease; need for ventilation for COVID-19 (defined as needing intubation or ECMO); and mortality (in-hospital and before-hospital) from COVID-19 disease. Each outcome was assessed independently, not as a composite outcome [27]. Additionally, back-pain was assessed as a negative control outcome [28]. Back pain was captured using concepts from various vocabularies, including CPT4, HCPCS, ICD10, ICD10CM, SNOMED, and Nebraska Lexicon, to capture outpatient diagnoses related to back pain and its synonyms.

**Table 1. Demographic and clinical characteristics of adults with Type 2 Diabetes treated by oral monotherapy and SARS-CoV-2 infection.**

| n (%) | | Overall (n = 6,626) | Metformin n = 5,236 | DPP4 inhibitors n = 459 | Sulfonylureas n = 931 | SMD** |
|---|---|---|---|---|---|---|
| Age, mean (SD) | | 60.69 (11.9) | 60.04 (11.9) | 62.50 (11.6) | 63.43 (11.5) | 0.193 |
| Age | 30–49 | 1,225 (18.5) | 1,035 (19.8) | 60 (13.1) | 131 (14.1) | 0.213 |
| | 50–59 | 1,732 (26.1) | 1,409 (26.9) | 121 (26.4) | 202 (21.7) | |
| | 60–69 | 2,039 (30.8) | 1,604 (30.6) | 142 (30.9) | 293 (31.5) | |
| | 70–79 | 1,289 (19.5) | 952 (18.2) | 104 (22.7) | 233 (25.0) | |
| | 80–85 | 341 (5.1) | 237 (4.5) | 32 (7.0) | 72 (7.7) | |
| Male | | 3,228 (48.7) | 2,546 (48.6) | 219 (47.7) | 463 (49.7) | 0.027 |
| Race, Ethnicity | White | 3,815 (57.6) | 2,988 (57.1) | 264 (57.5) | 563 (60.5) | 0.120 |
| | Black | 1,451 (21.9) | 1,121 (21.4) | 119 (25.9) | 211 (22.7) | |
| | Asian | 235 (3.5) | 192 (3.7) | <20 | 27 (2.9) | |
| | Hispanic/Latinx | 1,108 (16.7) | 925 (17.7) | 58 (12.6) | 125 (13.4) | |
| | Other/Unknown | 1,125 (17.0) | 935 (17.9) | <60 | 130 (14.0) | |
| BMI, mean (SD), kg/m$^2$ | | 34.09 (7.80) | 34.27 (7.85) | 33.07 (7.67) | 33.58 (7.55) | 0.104 |
| BMI Category (kg/m$^2$) | < = 25.0 | 527 (8.0) | 389 (7.4) | 56 (12.2) | 82 (8.8) | 0.14 |
| | 25.0 < 30.0 | 1,592 (24.0) | 1,232 (23.5) | 112 (24.4) | 248 (26.6) | |
| | 30.0 < 35.0 | 1,846 (27.9) | 1,475 (28.2) | 127 (27.7) | 244 (26.2) | |
| | 35.0 < 40.0 | 1,266 (19.1) | 1,012 (19.3) | 76 (16.6) | 178 (19.1) | |
| | > = 40.0 | 1,216 (18.4) | 987 (18.9) | 74 (16.1) | 155 (16.6) | |
| **Comorbidities, n (%)** | | | | | | |
| Heart failure | | 615 (9.3) | 469 (9.0) | 50 (10.9) | 96 (10.3) | 0.043 |
| Coronary Artery Disease | | 1,103 (16.6) | 830 (15.9) | 91 (19.8) | 182 (19.5) | 0.069 |
| Hypertension | | 5,111(77.1) | 3,997 (76.3) | 350 (76.3) | 764 (82.1) | 0.096 |
| COPD | | 496 (7.5) | 369 (7.0) | 46 (10.0) | 81 (8.7) | 0.071 |
| Cancer | | 755 (11.4) | 560 (10.7) | 70 (15.3) | 125 (13.4) | 0.091 |
| Liver disease | | 205 (3.1) | 155 (3.0) | < 20 | 39 (4.2) | 0.067 |
| CKD Stage 1, 2, or 3 | | 794 (12.0) | 525 (10.0) | 99 (21.6) | 170 (18.3) | 0.214 |
| Serum creatinine, mean (SD) | | 0.94 (0.32) | 0.92 (0.30) | 1.04 (0.40) | 1.00 (0.38) | 0.233 |
| **Medication prescriptions in the 90 days before +SARS-CoV-2 result** | | | | | | |
| ACEi | | 964 (14.5) | 1,091 (20.8) | 54 (11.8) | 141 (15.1) | 0.165 |
| ARB | | 1,286 (19.4) | 794 (15.2) | 48 (10.5) | 122 (13.1) | 0.094 |
| Statins | | 2,312 (34.9) | 1,942 (37.1) | 102 (22.2) | 268 (28.8) | 0.219 |
| Anti-coagulants | | 605 (9.1) | 484 (9.2) | 34 (7.4) | 87 (9.3) | 0.047 |
| Aspirin | | 579 (8.7) | 468 (8.9) | 36 (7.8) | 75 (8.1) | 0.026 |
| **Outcomes from Covid-19** | | | | | | |
| Hospitalization, ventilation, or mortality | | 17.9% | 17.0% | 22.0% | 20.8% | |
| Back pain (negative control) | | 1,689 (25.5) | 1,331 (25.4) | 238 (25.6) | 121 (26.4) | |

Abbreviations: DPP4i = dipeptidyl peptidase-4 inhibitor; PCR = polymerase chain reaction; SD = standard deviation; BMI = body mass index; SDM = standardized mean difference. COPD = chronic obstructive pulmonary disease; CKD = chronic kidney disease; ACEi = angiotensin converting enzyme inhibitor; ARB = angiotensin receptor blocker;

*p-value for differences between the 3 groups.

**SMD is the average of the 3 pairwise SMD's.

## Covariates

Potentially confounding covariates were identified based on clinical assessment of variables associated with the exposures and outcomes and are included in Table 1. Analysis was also adjusted for site, but this information is not included in Table 1. Comorbidities were defined

using translated OMOP concepts from ICD-10 codes in the previous 12 months. For chronic kidney disease, patients were additionally matched on serum creatinine (SCr) within the previous 12 months.

## Missingness

After excluding sites with greater than 90% missingness for BMI, weight was missing in 6.8%, height was missing in 8.5%, and serum creatinine level was missing in 17.2% of the cohort. With exception of weight, these missing data were addressed using the multiple imputation by chained equations (MICE) algorithm, where each incomplete variable is imputed stochastically by a separate model using fully conditional specification. After using MICE, BMI was missing in 2.7% of the overall cohort. All exposure, outcome, and confounder variables were included in the imputation models. The predictive mean matching method was used, with the passive imputation method used to specify deterministic dependencies among the columns, specifically BMI = weight/height$^2$ and the eGFR and creatinine, age, race, and gender relationship specified in the CKD-EPI eGFR equation [29]. Twenty completed data sets were constructed, the exposure and outcome models were fit to each data set separately as described below, and results were pooled using the Rubin method [30].

## Statistical analyses

For descriptive purposes, categorical variables were presented using counts and percentages, and continuous variables presented as means and standard deviation, for each exposure group. Differences among the 3 groups were summarized using the average Standardized Mean Difference (Table 1).

To adjust for confounding, we estimate weights with entropy balancing [31, 32]. Entropy balancing adjusts for confounding by exactly balancing means of confounders across treatment groups and can be viewed as an indirect approach of estimating the propensity score [33] but is empirically more robust [34]. In the balancing model, we include main effects in log BMI, sex, age, race, ethnicity, site, heart failure, coronary artery disease, chronic obstructive pulmonary disease, cancer, hypertension, liver disease, chronic kidney disease (stage 3 or lower), eGFR, past 90 days use of each of ARB, ACE inhibitor, statin, anticoagulant, and aspirin, and indicators for missing BMI and missing eGFR, and interactions between gender and hypertension, eGFR and statin, eGFR and gender, ACE and sex, eGFR and anticoagulant, eGFR and heart failure, and eGFR and COPD, as we observed substantial imbalances in these interactions. The outcome analysis proceeds in the same manner as an inverse probability of treatment weighting estimate. The summary of the balance between variables can be seen in Fig 2 (the 100 terms with the greatest imbalance before weighting), and Fig 3 (the 100 terms with the greatest imbalance after weighting). After weighting, the standardized absolute mean difference (SMD) was less than 0.05 for all terms.

These weights are then used in fitting a weighted relative risk regression model, which in addition to the balancing weights also includes covariates for all main effects in the balancing model, to construct doubly robust estimates of the relative effects of exposure on each outcome [35].

## Subgroup analyses

Prespecified subgroup analyses were conducted by sex and BMI based on previous literature.

Balance summaries, unweighted and weighted: 100 terms with the greatest imbalance after weighting

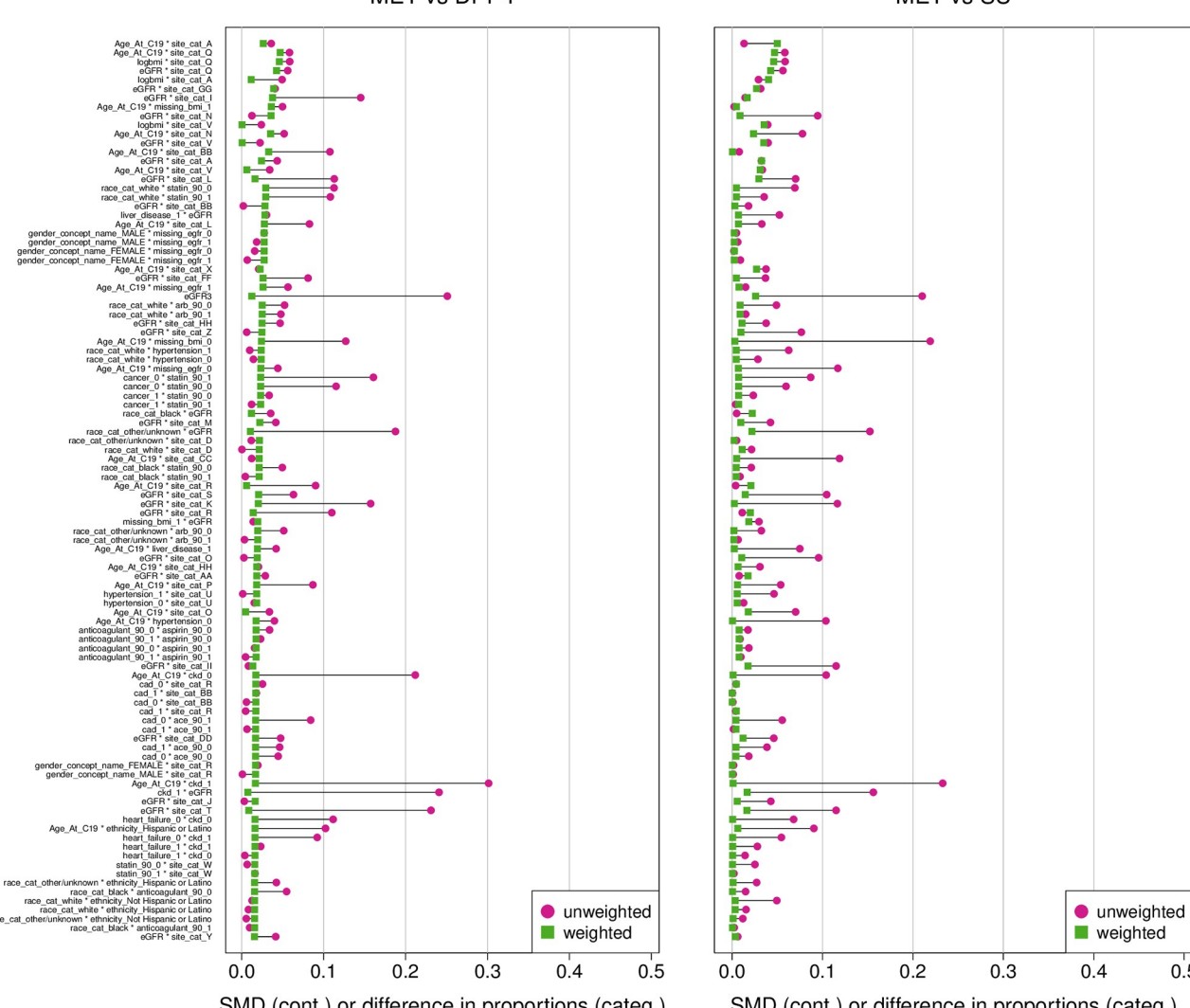

**Fig 2. The x axis the standardized mean difference for continuous variables and difference in proportions for categorical variables for the 100 terms with the greatest imbalance after weighing.** The circles represent the balance before weighting and the squares represent the balance after weighing.

## Sensitivity analyses

In order to understand whether selection bias caused us to misclassify individuals who had these chronic medications prescribed longer than 90 days before their +SARS-CoV-2 infection, we conducted sensitivity analyses using medications defined within the prior 180 and 270 days (S1 Table and S1, S2 Figs in S1 File). To assess for degree of unmeasured confounding that would be necessary to account for observed associations, we calculated e-values using the method outlined by VanderWeele et al [36]. Further sensitivity analyses will soon be possible in the data environment [37].

All analyses were conducted within the secure N3C computing environment using R statistical software (R Foundation for Statistical Computing, Vienna, Austria) including the

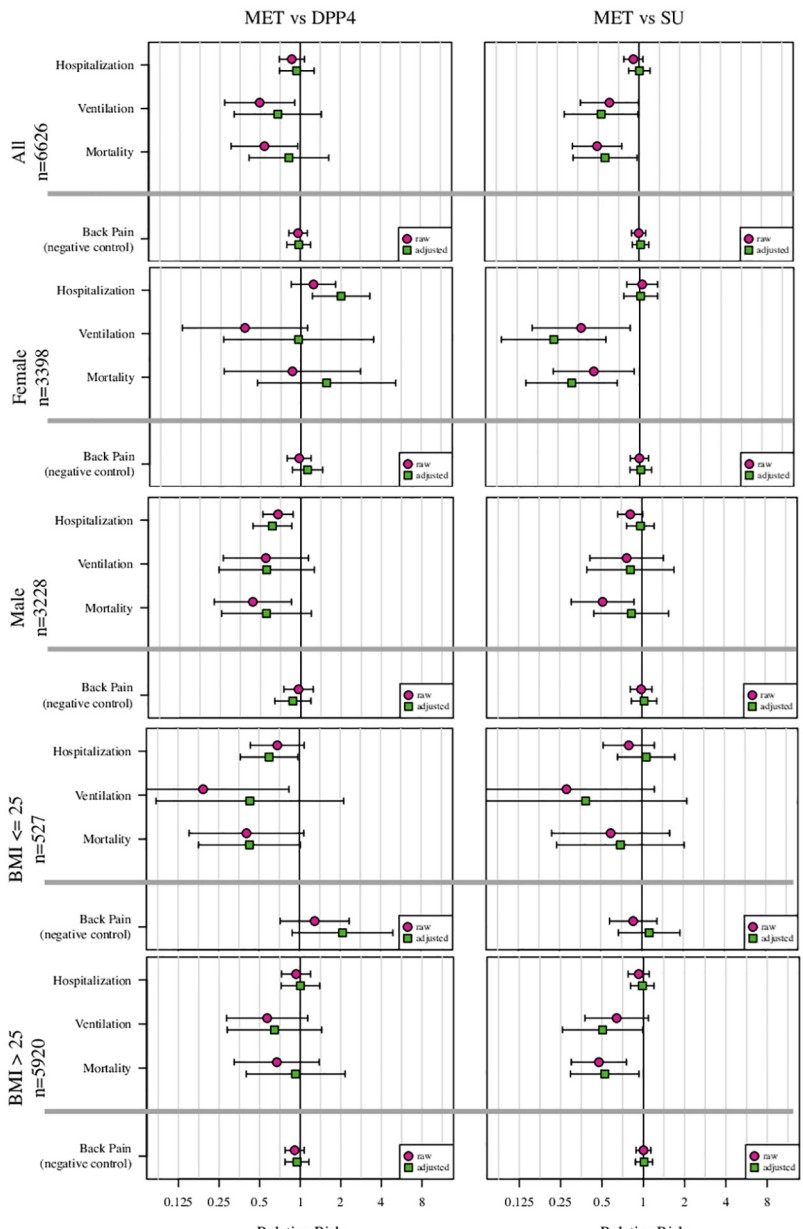

**Fig 3. This represents the risk ratios and 95% confidence intervals for the overall cohort as well as for subgroups by gender and subgroups by BMI ($<25$kg/m$^2$ and $\geq 25$kg/m$^2$).** The three panels on the left represent the metformin vs. DPP4i inhibitor comparison. The three panels on the right represent the metformin vs. SU comparison. The circles represent the raw comparison, and the squares represent the adjusted analysis. Within each panel, from top to bottom, the top result is risk of hospitalization; the 2nd result is risk of ventilation (including ECMO); the 3rd result is the risk of mortality; and the 4th comparison is the risk of a back pain, calculated as a negative control outcome. Abbreviations: MET = metformin; DPP4i = dipeptidyl peptidase 4 inhibitors; SU = sulfonylurea.

following packages: mice (for multiple imputation), [38] WeightIt and MatchThem (for estimating balancing weights), [39, 40] cobalt (for summarizing balance statistics), [41] survey (for inverse probability weighted regression), [42] and tableone (for descriptive statistics) [43]. Lastly, the statistical evidence of point/interval estimates were considered prior to relying on p-values' significance levels [44].

## Results

### Characteristics of the cohort

The total sample included 6,626 adults with T2D and positive SARS-CoV-2 test from 36 sites (Fig 1). The mean age was 60.7 +/- 12.0 years; 48.7% were male; 56.7% were White, 21.9% Black, 3.5% Asian, and 16.7% Latinx. The mean BMI was 34.1 +/- 7.8kg/m$^2$. Overall, 14.5% of the sample was hospitalized; 1.5% received mechanical ventilation; and 1.8% died.

The baseline demographic characteristics varied between the monotherapy cohorts: metformin users were younger than the SU and DPP4i users (60.0 versus 63.4 and 62.5 years, respectively). A greater percentage of metformin users were Latinx (17.7%) compared to SU (13.4%) and DPP4i (12.6%). The mean BMI in the metformin group was 34.3kg/m$^2$ compared to the SU (33.6) and DPP4i (33.1) groups. The DPP4i and SU groups had higher rates of cardiovascular disease, chronic renal disease, and cancer compared to the metformin group, Table 1.

In unadjusted frequencies, 14.2% of metformin uses were hospitalized, compared to 16.3% and 15.6% of DPP4i and SU users, respectively; 1.3% of metformin users were ventilated, compared to 2.6% and 2.1% of DPP4i and SU users, respectively; and 1.5% of metformin users died from COVID-19, compared to 2.8% and 3.1% of DPP4i and SU users, respectively.

The standardized mean difference between covariates before weighing ranged from approximately 0.05 to 0.50 (Supplement), and after weighting the SMD was < 0.05 for all covariates (Fig 2).

In adjusted outcomes metformin had non-significant associations with reduced severity of COVID-19 compared to DPP4i (Fig 3). Compared to SU, metformin was associated with a lower risk of mortality (RR 0.56, 95%CI 0.33–0.97, p = 0.037) and needing ventilation (RR 0.5, 95% CI 0.28–0.98, p = 0.044). There was no difference between the cohorts in unadjusted or adjusted results of the negative control outcome, back pain (Table 2, Fig 3).

For subgroup analyses, there was evidence that the treatment effect of metformin relative to SU on ventilation differed between females and males with a sex by treatment interaction p = 0.02; and on mortality, p = 0.05 (Table 2, Fig 3). There was no difference in outcomes between BMI subgroups. The sensitivity analyses using 180 and 270 days for capturing chronic medication use showed similar results (Supplement). The e-values for the adjusted model ranged from 1.11 to 8.16. E-values indicate the magnitude of association that an unmeasured confounder would need to have with both the treatment (or in the case of a RR<1 the control, either DPP4i or SU) and outcome, beyond the measured confounders, to account for any observed association.

## Discussion

This analysis of adults with T2DM and +SARS-CoV-2 infection was the first analysis of prevalent users of diabetes monotherapy and was possible because of the size of this database. We found that compared to SU use, metformin use was significantly associated with less severe outcomes from COVID-19 compared to SU users, but associations were not significant compared to DPP4i use. The size of this database allowed us to conduct this analysis with prevalent user comparator groups of diabetes medications that are therapeutically similar, as SU and DPP4i are less common than metformin. We feel this approach has advantages over a non-user comparison, as it explicitly compares to patients receiving an alternative treatment for the same indication, which is a significant consideration when assessing diabetes medications and outcomes from COVID-19 in persons with T2DM. A recent paper by Wang et al, [45] conducted a similar analysis in adults with T2DM comparing metformin to other diabetes medications. They found favorable hazard ratios for metformin compared to the other diabetes

**Table 2. Risk ratios for severe COVID-19 outcomes in the overall cohort as well as subgroups by gender.**

| Overall cohort, metformin versus DPP4i (n = 5,695*) | | | | Female (n = 2,930*) | | Male (n = 2,756*) | |
|---|---|---|---|---|---|---|---|
| Outcome | Model | RR (95% CI) | p value | RR (95% CI) | p value | RR (95% CI) | p value |
| Hospitalization | Crude | 0.87 (0.70–1.08) | 0.195 | 1.24 (0.85–1.82) | 0.27 | 0.61 (0.44–0.86) | <0.01 |
| *Number of events* | Adjusted | 0.94 (070–1.27) | 0.700 | 2.00 (1.22–3.29) | <0.01 | 0.68 (0.52–0.88) | <0.01 |
| | | 816 | | 373 | | 433 | |
| Ventilation | Crude | 0.50 (0.27–0.91) | 0.024 | 0.38 (0.13–1.12) | 0.08 | 0.55 (0.26–1.14) | 0.11 |
| *Number of events* | Adjusted | 0.68 (0.32–1.44) | 0.315 | 0.96 (0.26–3.50) | 0.95 | 0.56 (0.24–1.26) | 0.16 |
| | | 80 | | 21 | | 59 | |
| Mortality | Crude | 0.54 (0.30–0.96) | 0.036 | 0.86 (0.26–2.81) | 0.81 | 0.44 (0.23–0.85) | 0.02 |
| *Number of events* | Adjusted | 0.82 (0.41–1.64) | 0.581 | 1.56 (0.47–5.16) | 0.47 | 0.55 (0.26–1.20) | 0.13 |
| | | 93 | | 32 | | 61 | |
| Back pain | Crude | 0.96 (0.82–1.13) | 0.656 | 0.97 (0.79–1.19) | 0.77 | 0.96 (0.75–1.24) | 0.76 |
| (negative control) | Adjusted | 0.98 (0.79–1.20) | 0.816 | 1.12 (0.86–1.45) | 0.84 | 0.87 (0.64–1.19) | 0.39 |
| *Number of events* | | 1,452 | | 831 | | 621 | |
| Overall cohort, metformin versus SU (n = 6,167**) | | | | Female (n = 3,158**) | | Male (3,009**) | |
| Outcome | Model | RR (95% CI) | p value | RR (95% CI) | p value | RR (95% CI) | p value |
| Hospitalization | Crude | 0.91 (0.77–1.07) | 0.252 | 1.04 (0.80–1.35) | 0.75 | 0.82 (0.67–1.01) | 0.07 |
| *Number of events* | Adjusted | 1.01 (0.84–1.21) | 0.945 | 1.02 (0.76–1.35) | 0.91 | 0.98 (0.77–1.23) | 0.84 |
| | | 886 | | 406 | | 480 | |
| Ventilation | Crude | 0.60 (037–0.99) | 0.046 | 0.37 (0.16–0.85) | 0.02 | 0.77 (0.42–1.44) | 0.42 |
| *Number of events* | Adjusted | 0.53 (0.28–0.98) | 0.044 | 0.23 (0.09–0.56) | <0.01 | 0.82 (0.39–1.72) | 0.61 |
| | | 88 | | 25 | | 63 | |
| Mortality | Crude | 0.49 (0.32–0.75) | 0.001 | 0.46 (0.23–0.91) | 0.03 | 0.52 (0.30–0.87) | 0.01 |
| *Number of events* | Adjusted | 0.56 (0.33–0.97) | 0.037 | 0.31 (0.14–0.68) | <0.01 | 0.84 (0.44–1.57) | 0.58 |
| | | 109 | | 40 | | 69 | |
| Back pain | Crude | 0.99 (0.88–1.12) | 0.926 | 1.00 (0.85–1.16) | 0.95 | 0.99 (0.82–1.19) | 0.89 |
| (negative control) | Adjusted | 1.03 (0.89–1.18) | 0.710 | 1.02 (0.84–1.22) | 0.84 | 1.04 (0.84–1.29) | 0.74 |
| *Number of events* | | 1,569 | | 894 | | 675 | |

Abbreviations: DPP4i = dipeptidyl peptidase-4 inhibitors; SU = sulfonylureas.

*n is for metformin + DPP4i users;

**n is for metformin + SU users.

medications, but none of the matched analyses reached the 5% level of statistical significance [44].

We conducted a prespecified subgroup analysis by sex based on earlier work showing that metformin lowers CRP more in women than men, improved cancer mortality in women but not men, and conveyed greater protection against severe outcomes from COVID-19 in women compared to men [46]. The association with lower risk of ventilation and mortality with metformin versus SU was significant for females but not for males in this analysis. This potential influence of sex as a biologic variable should be further assessed. Much of the mechanistic research on metformin and DPP4i's was done before 2014, when the NIH started to promote the study of sex as a biologic variable [47]. However metformin has been found to reduce TNF-alpha, IL-6, and possibly boost IL-10 in females more than males, which is relevant to the pathophysiology of COVID-19 [48–50].

Subgroup analysis was conducted comparing those with a BMI>25kg/m$^2$ (the definition of overweight, and the BMI at which visceral adiposity starts to accumulate more rapidly) to those with a BMI<25kg/m$^2$ [51]. If metformin were effective only in individuals with an

elevated BMI, the antiviral actions of metformin might be less significant than anti-inflammatory and anti-thrombotic effects of metformin. However, we saw no obvious difference between these BMI groups. It is possible that this BMI threshold is too low, or that potential benefit from metformin is not dependent on baseline amount of adipokines (many of which are associated with poor outcomes from COVID-19).

These results may contribute to the growing body of evidence suggesting that metformin use may be associated with less severe COVID-19 disease. There is also in-silico, in-vitro, and in-tissue data suggesting that metformin associated with less severe outcomes from COVID-19 [10, 11, 16–18]. Metformin is safe in nearly all individuals, including individuals with heart, liver, and kidney disease, but should be used with caution in persons with advanced heart, liver, or kidney disease [9, 52–56]. Metformin has very few interactions with other medications and requires no follow-up until after 1 year of use, making it an ideal option for persons on other chronic medications or persons with lack of access to follow-up care.

Given the significant global impact of SARS-CoV-2 and the COVID-19 pandemic, patients should have several options for safe, available, inexpensive early outpatient treatment of SARS-CoV-2 infection to prevent severe COVID-19 disease. There is also evidence that early outpatient treatment with may possibly prevent long COVID symptoms (post-acute sequelae of COVID, PASC) [57].

While in-vitro and in-silico data supports its use in active infection, observational analyses such as this only add information about metformin use before infection with SARS-CoV-2. Few papers describe metformin continued or initiated during hospitalizations for COVID [58]. Randomized trials are needed to understand whether metformin has any efficacy in the setting of SARS-CoV-2 infection, exposure to infection, or treatment and prevention of PASC. Metformin's safety and cost make it a medication that is low-risk enough to reasonably consider using in a PEP fashion. While viral variants may evade vaccine-induced immunity because of their cell-entry abilities, they will still depend on host proteins for transcription and translation. Metformin's inhibition of proteins that are critical to viral replication may mean it is still relevant for most viral variants.

## Limitations

This observational analysis is subject to residual unmeasured confounding and bias. The degree of confounding typically seen in the assessment of repurposed medications for outpatient treatment of COVID-19 is not yet well established and in our setting with an active comparator, we would generally assume associations of an unmeasured confounder with treatment to be smaller than associations with the outcome. Because of sample size limitations, we are not able to perform the analysis using a new user active comparator design which may lead to a variety of biases [45]. In order to reduce ascertainment and misclassification bias, analyses were restricted to persons with at least one outpatient healthcare encounter in the previous 12 months, and prescriptions from the previous 90 days [59]. Records of individuals over age 85 were excluded to reduce confounding by frailty, and persons with CKD stages 4, 5, and ESRD were excluded to reduce confounding by contraindication [45]. It is not known whether the persons in these cohorts continued their metformin, SU, and DPP4i use during their SARS-CoV-2 infection. Given that there are several hypotheses as to how metformin might reduce severity of COVID-19 disease, it is not known if use prior to infection, during infection, or after initial acute infection is associated with the results observed in this analysis, and the associations may not generalize beyond adults with type 2 diabetes.

## Conclusions

In this retrospective cohort analysis of adults with T2DM and COVID-19 in a large, geographically diverse dataset there were statistically significant associations between metformin use and less severe outcomes from COVID-19 compared to SU use, but not compared to DPP4i use. Due to the size of the database, this was the first analysis able to compare outcomes across diabetes monotherapy groups, so this manuscript has methodologic strengths over previous observational analyses. This analysis adds to the literature suggesting a potential role for metformin in early treatment and possible post-exposure prophylaxis for COVID-19 disease, but we could not specifically address this hypothesis. Early outpatient treatment with safe and available therapeutics is particularly important for areas of the world with limited access to the vaccines and other COVID-19 therapies. New user cohort studies are needed, but the number of persons initiating oral T2DM treatment during acute SARS-CoV-2 infection may be small. Randomized trials of early outpatient treatment are needed and underway, and randomized trials of post-exposure prophylaxis are also needed.

## Supporting information

**S1 File.**
(DOCX)

## Acknowledgments

Consortial Authors: Melissa A. Haendel, PhD; Christopher G. Chute, MD, Dr.P.H, M.P.H.

This research was possible because of the patients whose information is included within the data and the organizations (https://ncats.nih.gov/n3c/resources/data-contribution/data-transfer-agreement-signatories) and scientists who have contributed to the on-going development of this community resource [25].

## Author Contributions

**Conceptualization:** Victor Garcia, Jeremy Harper, John B. Buse.

**Data curation:** Steven G. Johnson, Victor Garcia, Jeremy Harper, Stephenie Hong, Joy Alamgir, Trine Abrahamsen.

**Formal analysis:** Michael D. Evans, Kaifeng Yang.

**Investigation:** Carolyn T. Bramante, Jena Tronieri, Anna Kahkoska, Farrukh Koraishy, John B. Buse.

**Methodology:** Kenneth J. Wilkins, Jared D. Huling, Hemalkumar Mehta, Caleb Alexander, Jena Tronieri, Til Stürmer, John B. Buse.

**Project administration:** Carolyn T. Bramante.

**Resources:** Carolyn T. Bramante.

**Supervision:** Carolyn T. Bramante, John B. Buse.

**Validation:** Victor Garcia.

**Writing – original draft:** Carolyn T. Bramante, Katrina Hartman, Til Stürmer.

**Writing – review & editing:** Carolyn T. Bramante, John B. Buse.

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
