## [Decision Letter · Decision Letter 0]

11 Feb 2022

PONE-D-21-35574Diabetes medications and Associations with Covid-19 Outcomes in the N3C Database: A National Retrospective Cohort StudyPLOS ONE

Dear Dr. Carolyn T Bramante,

Thank you for submitting your manuscript to PLOS ONE. After careful consideration, we feel that it has merit but does not fully meet PLOS ONE’s publication criteria as it currently stands. Therefore, we invite you to submit a revised version of the manuscript that addresses the points raised during the review process.

We look forward to receiving your revised manuscript.

Kind regards,

Surasak Saokaew, PharmD, RPh, PhD, BPHCP, FACP, FCPA

Academic Editor

PLOS ONE

Journal Requirements:

“The analyses described in this publication were conducted with data or tools accessed through the National Institutes of Health’s National Center for Advancing Translational Sciences (NCATS) N3C Data Enclave (https://covid.cd2h.org) and N3C Attribution & Publication Policy v 1.2-2020-08-25b supported by NCATS U24 TR002306 and NCATS grants KL2TR002492 (UMN) and UL1TR002494 (UMN) and UL1TR002489 (UNC). The content is solely the responsibility of the authors and does not necessarily represent the official views of the National Institutes of Health’s National Center for Advancing Translational Sciences. TS receives investigator-initiated research funding and support as Principal Investigator (R01 AG056479) from the National Institute on Aging (NIA), and as Co-Investigator (R01 HL118255, R01MD011680), National Institutes of Health (NIH). He also receives salary support as Director of Comparative Effectiveness Research (CER), NC TraCS Institute, UNC Clinical and Translational Science Award (UL1TR002489), the Center for Pharmacoepidemiology (current members: GlaxoSmithKline, UCB BioSciences, Takeda, AbbVie, Boehringer Ingelheim), from pharmaceutical companies (Novo Nordisk), and from a generous contribution from Dr. Nancy A. Dreyer to the Department of Epidemiology, University of North Carolina at Chapel Hill. Dr. Stürmer does not accept personal compensation of any kind from any pharmaceutical company.”

“The analyses described in this publication were conducted with data or tools accessed through the National Institutes of Health’s National Center for Advancing Translational Sciences (NCATS) N3C Data Enclave (https://covid.cd2h.org) and N3C Attribution & Publication Policy v 1.2-2020-08-25b supported by NCATS U24 TR002306 and NCATS grants KL2TR002492 (UMN) and UL1TR002494 (UMN) and UL1TR002489 (UNC). The content is solely the responsibility of the authors and does not necessarily represent the official views of the National Institutes of Health’s National Center for Advancing Translational Sciences. TS receives investigator-initiated research funding and support as Principal Investigator (R01 AG056479) from the National Institute on Aging (NIA), and as Co-Investigator (R01 HL118255, R01MD011680), National Institutes of Health (NIH). He also receives salary support as Director of Comparative Effectiveness Research (CER), NC TraCS Institute, UNC Clinical and Translational Science Award (UL1TR002489), the Center for Pharmacoepidemiology (current members: GlaxoSmithKline, UCB BioSciences, Takeda, AbbVie, Boehringer Ingelheim), from pharmaceutical companies (Novo Nordisk), and from a generous contribution from Dr. Nancy A. Dreyer to the Department of Epidemiology, University of North Carolina at Chapel Hill. Dr. Stürmer does not accept personal compensation of any kind from any pharmaceutical company.”

“The analyses described in this publication were conducted with data or tools accessed through the National Institutes of Health’s National Center for Advancing Translational Sciences (NCATS) N3C Data Enclave (https://covid.cd2h.org) and N3C Attribution & Publication Policy v 1.2-2020-08-25b supported by NCATS U24 TR002306 and NCATS grants KL2TR002492 (UMN) and UL1TR002494 (UMN) and UL1TR002489 (UNC). The content is solely the responsibility of the authors and does not necessarily represent the official views of the National Institutes of Health’s National Center for Advancing Translational Sciences. TS receives investigator-initiated research funding and support as Principal Investigator (R01 AG056479) from the National Institute on Aging (NIA), and as Co-Investigator (R01 HL118255, R01MD011680), National Institutes of Health (NIH). He also receives salary support as Director of Comparative Effectiveness Research (CER), NC TraCS Institute, UNC Clinical and Translational Science Award (UL1TR002489), the Center for Pharmacoepidemiology (current members: GlaxoSmithKline, UCB BioSciences, Takeda, AbbVie, Boehringer Ingelheim), from pharmaceutical companies (Novo Nordisk), and from a generous contribution from Dr. Nancy A. Dreyer to the Department of Epidemiology, University of North Carolina at Chapel Hill. Dr. Stürmer does not accept personal compensation of any kind from any pharmaceutical company.”

“I have read the journal's policy and the authors of this manuscript have the following competing interests: TS owns stock in Novartis, Roche, and Novo Nordisk. T.J.A. is an employee of Novo Nordisk and reports personal fees and non-financial support from Novo Nordisk during the conduct of the study, as well as personal fees from Novo Nordisk outside the submitted work. J.A. is founder of ARIScience.

J.B.B.’s contracted consulting fees and travel support for contracted activities are paid to the University of North Carolina by Adocia, AstraZeneca, Eli Lilly, Intarcia Therapeutics, MannKind, Novo Nordisk, Sanofi, Senseonics, and vTv Therapeutics;  he reports grant support from AstraZeneca, Dexcom, Eli Lilly, Intarcia Therapeutics, Johnson & Johnson, Lexicon, NovaTarg, Novo Nordisk, Sanofi, Theracos, Tolerion, and vTv Therapeutics; he has received fees for consultation from Anji Pharmaceuticals, AstraZeneca, Boehringer Ingelheim, Cirius Therapeutics Inc, Eli Lilly, Fortress Biotech, Janssen, Mellitus Health, Moderna, Pendulum Therapeutics, Praetego, Stability Health, and Zealand Pharma; he holds stock/options in Mellitus Health, Pendulum Therapeutics, PhaseBio, Praetego, and Stability Health; and he is supported by grants from the National Institutes of Health, Patient Centered Outcomes Research Institute, Juvenile Diabetes Research Foundation International and the American Diabetes Association. Dr. Bramante owns an FDA IND for investigation of metformin for early outpatient treatment of COVID-19, NCT04510194.”

Reviewers' comments:

Reviewer's Responses to Questions

**Comments to the Author**

1. Is the manuscript technically sound, and do the data support the conclusions?

Reviewer #1: Yes

Reviewer #2: Yes

2. Has the statistical analysis been performed appropriately and rigorously? 

Reviewer #1: Yes

Reviewer #2: I Don't Know

3. Have the authors made all data underlying the findings in their manuscript fully available?

Reviewer #1: Yes

Reviewer #2: Yes

4. Is the manuscript presented in an intelligible fashion and written in standard English?

Reviewer #1: Yes

Reviewer #2: Yes

5. Review Comments to the Author

Reviewer #1: Thank you for the opportunity to review this manuscript. The authors present interesting finding on metformin and associations with Covid-19 outcomes in a National databases. However, some issues should be addressed before acceptance for publication.

Introduction

1. This study presents the results of original research. However, I find new evidence has shown that metformin favorably influences COVID-19 outcomes. For example in “Metformin in Patients With COVID-19: A Systematic Review and Meta-Analysis” https://doi.org/10.3389/fmed.2021.704666. So I think it will be good if authors give more details on what kind of differences you made from previous studies.

2. Because DPP4i have also been hypothesized to reduce severity of Covid-19 disease, so comparing individuals with type 2 DM treated with monotherapy is make sense. On the other hand, this research compared metformin to SU, which have no hypothesized benefit in Covid-19 patients. Could you explain more why the authors hypothesized metformin use would be associated with less severe Covid-19 outcome than SU and DPP4i use?

Methods

3. Base on the design and population, why do the authors want EHR data from 2017 to 2021 rather than 2019/2020 to 2021.

4. It's possible that the inclusion criteria will need to be clarified. Why was the study limited to those over the age of 30? Is it true that they haven't been vaccinated yet?

Patients were excluded from the analysis if any other diabetes medication were list in the 90 days prior to the SARs cov 2 result.

These variables may have an impact on the study's outcome.

Base on this comment, there should be further discussed. What populations can this study's findings be applied to?

5. To adjust for confounding, authors included many factors in the model. How were the factors selected? Please explain. I also notice that the authors' analytical models did not some incorporate factors that impact COVID severity and mortality, such as smoking.

Discussion

6. Beyond reporting the findings, I would appreciate if the authors explain whether the results of this study should be applied to patients with diabetes who regularly use more than 1 diabetic medicines, and if so, how?

Reviewer #2: Comments

-The study aim is missing in the abstract.

-Methods and Results in the abstract should be revised and rearranged.

-Table 1, “Hospitalization, ventilation, or mortality” raw: in addition to percent number in each group should be added and p_value should be included.

6. PLOS authors have the option to publish the peer review history of their article (what does this mean?). If published, this will include your full peer review and any attached files.

Reviewer #1: **Yes: **Adinat Umnuaypornlert

Reviewer #2: No

---

## [Author Response · Author response to Decision Letter 0]

21 Apr 2022

Thank you for the opportunity to revise this manuscript, addressing the reviewers' comments has made it a stronger manuscript. 

Reviewer #1: Thank you for the opportunity to review this manuscript. The authors present interesting finding on metformin and associations with Covid-19 outcomes in a National databases. However, some issues should be addressed before acceptance for publication.

Introduction 

1. This study presents the results of original research. However, I find new evidence has shown that metformin favorably influences COVID-19 outcomes. For example in “Metformin in Patients With COVID-19: A Systematic Review and Meta-Analysis” https://doi.org/10.3389/fmed.2021.704666. So I think it will be good if authors give more details on what kind of differences you made from previous studies.

Thank you for this comment. Our initial submission to Plos medicine was actually in August 2021 around when that systematic review was published. We added text to the introduction to try to better justify why we feel this analysis is valuable (Lines 122 – 126), and re-organized paragraph 3 of the introduction to further clarify, and discussion (292-293). 

2. Because DPP4i have also been hypothesized to reduce severity of Covid-19 disease, so comparing individuals with type 2 DM treated with monotherapy is make sense. On the other hand, this research compared metformin to SU, which have no hypothesized benefit in Covid-19 patients. Could you explain more why the authors hypothesized metformin use would be associated with less severe Covid-19 outcome than SU and DPP4i use?

Thank you, we think that adding clarifications on these points makes the manuscript stronger. They have been added to: Lines 122-126; 136-138; 140-141, 144. 

Methods Base on the design and population, why do the authors want EHR data from 2017 to 2021 rather than 2019/2020 to 2021.

The database includes data from 2017, which would include the EHR coding for diabetes which is valuable for defining the cohorts but does not offer information about Covid. 

It's possible that the inclusion criteria will need to be clarified. Why was the study limited to those over the age of 30? Is it true that they haven't been vaccinated yet?

Thank you, we added clarification to line 173, and thank you we added clarification about the end date of the cohort (it was before the vaccine was available). 

Patients were excluded from the analysis if any other diabetes medication were list in the 90 days prior to the SARs cov 2 result. These variables may have an impact on the study's outcome. Base on this comment, there should be further discussed. What populations can this study's findings be applied to?

Thank you, yes because we were trying to isolate the effects of metformin’s potential actions in COVID-19 disease compared to its diabetes treatment actions, we did not include other diabetes medications that could confound the association. Strictly speaking, the findings are only an association and should likely not guide clinical care, but we added text to the limitations, line 355. 

5. To adjust for confounding, authors included many factors in the model. How were the

factors selected? Please explain. I also notice that the authors' analytical models did not 

some incorporate factors that impact COVID severity and mortality, such as smoking.

The factors were selected using a conceptual model of potential confounders related to the exposure and outcome, and then balanced them across the exposure cohorts. We did not use a machine learning or otherwise automated approach to select factors for adjustment. 

Discussion

6. Beyond reporting the findings, I would appreciate if the authors explain whether the results of this study should be applied to patients with diabetes who regularly use more than 1 diabetic medicines, and if so, how?

Thank you, we added text to address this good point about what this analysis adds, to lines 355; 359-361; and lines 292 to 293, which should set readers up for the rest of the first paragraph of the discussion. 

Reviewer 2: Table 1, “Hospitalization, ventilation, or mortality” raw: in addition to percent number in each group should be added and p_value should be included.

Thank you for this point. We did not have a specific hypothesis about that comparison, thus adding a p value would not be an appropriate use of the data given our data use agreement. This view is in concordance with guidance from the American Statistical Association on the use of p values. We will add citations to this effect to the methods: american statistical association point about p value here. https://www.amstat.org/asa/files/pdfs/p-valuestatement.pdf

Sincerely,

Carolyn Bramante, MD MPH

Assistant Professor of Medicine

University of Minnesota

---

## [Decision Letter · Decision Letter 1]

4 Jul 2022

Diabetes medications and Associations with Covid-19 Outcomes in the N3C Database: A National Retrospective Cohort Study

PONE-D-21-35574R1

Dear Dr. Carolyn T Bramante,

We’re pleased to inform you that your manuscript has been judged scientifically suitable for publication and will be formally accepted for publication once it meets all outstanding technical requirements.

Kind regards,

Surasak Saokaew, PharmD, RPh, PhD, BPHCP, FACP, FCPA

Academic Editor

PLOS ONE

Additional Editor Comments (optional):

Reviewers' comments:

Reviewer's Responses to Questions

**Comments to the Author**

1. If the authors have adequately addressed your comments raised in a previous round of review and you feel that this manuscript is now acceptable for publication, you may indicate that here to bypass the “Comments to the Author” section, enter your conflict of interest statement in the “Confidential to Editor” section, and submit your "Accept" recommendation.

Reviewer #2: All comments have been addressed

2. Is the manuscript technically sound, and do the data support the conclusions?

Reviewer #2: Yes

3. Has the statistical analysis been performed appropriately and rigorously? 

Reviewer #2: Yes

4. Have the authors made all data underlying the findings in their manuscript fully available?

Reviewer #2: Yes

5. Is the manuscript presented in an intelligible fashion and written in standard English?

Reviewer #2: Yes

6. Review Comments to the Author

Reviewer #2: There is no comment

The author has provided proper response to the comments and revised the manuscript

7. PLOS authors have the option to publish the peer review history of their article (what does this mean?). If published, this will include your full peer review and any attached files.

Reviewer #2: No

---

## [Editor Report · Acceptance letter]

3 Nov 2022

PONE-D-21-35574R1 

Diabetes medications and Associations with Covid-19 Outcomes in the N3C Database: A National Retrospective Cohort Study 

Dear Dr. Bramante:

I'm pleased to inform you that your manuscript has been deemed suitable for publication in PLOS ONE. Congratulations! Your manuscript is now with our production department. 

Kind regards, 

on behalf of

Dr. Surasak Saokaew 

Academic Editor

PLOS ONE